# Fishways as Downstream Routes in Small Hydropower Plants: Experiences with a Potamodromous Cyprinid



**Francisco Javier Sanz-Ronda [1,*], Juan Francisco Fuentes-Pérez [1,2], Ana García-Vega [1] and Francisco Javier Bravo-Córdoba [1,2]**

1   GEA-Group of Applied Ecohydraulics, Department of Hydraulics and Hydrology, University of Valladolid, Avda. Madrid 44, 34004 Palencia, Spain; jfuentes@iaf.uva.es (J.F.F.-P.); ana.garcia.vega@iaf.uva.es (A.G.-V.); francisco.bravo@iaf.uva.es (F.J.B.-C.)
2   Centro Tecnológico Agrario y Agroalimentario, Itagra.ct. Avda. Madrid 44, 34004 Palencia, Spain
*   Correspondence: jsanz@uva.es; Tel.: +34-979108358

**Abstract:** Fish need to move upstream and downstream through rivers to complete their life cycles. Despite the fact that fishways are the most commonly applied solution to recover longitudinal connectivity, they are not considered viable for downstream migration. Therefore, alternative facilities are recommended to facilitate downstream migration. However, a few recent studies have disagreed with this general assumption, showing the potential for bidirectional movements. This study advances our understanding of the potential of fishways for downstream migration by studying their efficiency in a run-of-the-river hydropower plant in the Duero River (Spain). To achieve this, downstream movements of the Iberian barbel ($n$ = 299) were monitored in a stepped fishway for two years with passive integrated transponder (PIT)-tag technology, considering the effect of fish origin and release zone. The results showed that 24.9% of barbels descended through the fishway, with the origin and release zone affecting the fishway location. In addition, downstream movements were observed throughout the whole year, except in winter. The study concludes that, under specific scenarios, fishways could act as safe alternative routes for downstream migration.

**Keywords:** downstream migration; bidirectional movement; fish passes; potamodromous fish; Iberian barbel



## 1. Introduction

Many fish species display migratory movements between different habitats as a strategy to optimize their fitness and complete their life cycles [1,2]. The nature and extent of these displacements differ in space and time. It is possible to find fish species that complete their life cycles in marine environments (oceanodromous), strictly riverine species (potamodromous), and even species that share different stages between seas and rivers (diadromous) [2]. Moreover, these movements may differ between individuals in a population (partial migration), with the existence of some fish that only develop short movements (resident fish) and others that complete long-distance movements (mobile fish) [3–5]. Migratory behavior is a complex and variable phenomenon which shows differences between species, life stages, individuals, and regions [2].

Rheophilic potamodromous cyprinids, such as barbels (genus *Luciobarbus* and *Barbus*), perform movements during most of the year, with the greatest migratory activity taking place during spring and early summer. During this period they ascend to the headwaters for reproduction [6,7], looking for shallow waters with high oxygen concentrations, gravel and pebble bottoms, and a moderate current velocity [8,9]. Within a few days (females) to several weeks (males) after spawning, barbels move to deeper waters, searching for summer refuge habitats [5,10–12], and then, they reduce their mobility [13]. From late summer to autumn, barbels move to overwinter places [14,15], where they enter a dormancy period during winter (i.e., without activity) as a consequence of the low temperatures [6,16].

Barbels exhibit partial migration, with both resident and mobile individuals being present in the same population. Resident fish show a shorter home range (up to 1 km) and fidelity to spawning and summer/winter refuge sites. Alternatively, mobile individuals display a larger home range (more than 20 km) and low fidelity to seasonal habitats [17,18]. Upstream movements are usually, but not always, related to the pre-spawning season, whereas downstream movements are related to post-spawning and the search for summer/winter habitats [5,6]. However, a spatiotemporal variability is expected, depending on the river position and environmental conditions [2,19].

Therefore, considering the mobility requirements of fish, bidirectional river connectivity is essential in order to ensure that they can complete their life cycles [20,21]. However, many obstacles (e.g., dams, weirs, culverts, etc.) hinder fish movement and block their migration routes, causing direct impacts on fish population fitness and endangering their conservation [22]. Fishways, such as pool-type, baffle-type, rocky ramps, nature-like bypass channels, locks, lifts, etc., are the most commonly used devices to recover upstream longitudinal connectivity [23–25]. In the case of downstream connectivity, there are also specific devices for bypassing obstacles and protect migratory fish [26,27], such as barriers (physical, mechanical, and/or behavioral) that hinder entry to hazardous facilities (turbines or diversion channels), or guiding structures to safety bypasses (e.g., sluices) [27,28]. Despite the existence of downstream migration aids, many water managers and hydropower operators assume that spillways or turbines are effective downstream migration routes [26,28–30], thus, downstream movement is frequently devalued [29], especially for non-diadromous species. In the case of Spain, river connectivity projects that consider downstream migration measures are unusual and mainly target diadromous fish (e.g., IREKIBAI (LIFE 14 NAT/ES/00186; www.irekibai.eu, accessed on 10 March 2021)).

Rheophilic fish follow and drift with the main flow during their downstream migration [26,28,31]. Their preferred corridors are waterways with the highest flow volume [29]. According to this behavior, in the case of hydropower plants (HPP), fish tend to be attracted to areas of high flow, such as those near spillways or the intake of the turbines [32]. On the one hand, if the total river discharge is abstracted and the dam/weir does not overflow, the only option to go downward is entering the diversion channels (if they exist) or through the turbines [26], with an associated probability of injury or mortality [33]. Mortality may be immediate or delayed (e.g., cuts, strikes, barotrauma), or may even affect the fish indirectly (e.g., susceptibility to predation), and it varies with turbine design and operation [34,35]. When the dam/weir is overflowing, fish prefer to approach spillways [28,32,36] as these are usually a safer path. On the other hand, fishways could be a possible route for downstream movements, but their viability as downstream routes is considered nil or minimal in large rivers and dams [30,37,38], although some studies show effective downward passage [39,40].

Statistically, small dams (associated with small HPP facilities) and weirs (usually for irrigation) contribute more to the disruption of river connectivity than large dams [22,41,42]. In general, small HPPs (considered as those with installed capacity up to 10 MW in the EU) present a run-of-the-river scheme installation [43]. These facilities make use of the entire river flow except for those to guarantee the fishway operation (when it exists). Thus, spillway overflow is not common, reducing its availability as a route for the descent only during high-flow events. Moreover, in Mediterranean climates, high-flow events are scarce and concentrated outside the downstream movement period of cyprinids (due to summer droughts, saving water for irrigation, consumption, etc.) [44,45]. In these cases, and the absence of specific downstream migration devices, the intake of the HPP and fishways are the only possible routes. Therefore, it is crucial to determine the effectiveness and to understand the performance of fishways as possible downstream passage routes, to confirm or refute safe bidirectional connectivity in these common HPP configurations.

Considering the above, this study analyzes the performance of a fishway located in a run-of-the-river hydropower plant as a possible route for downstream fish movement. The studied facility is located in the Duero River (Spain) and the target species was the

Iberian barbel (*Luciobarbus bocagei*, Steindachner, 1864) (hereafter referred to as barbel), a representative species of several medium-sized potamodromous cyprinids in the Iberian Peninsula and circum-Mediterranean region [46]. The main objectives of the study were to (1) assess the fishway as a downstream migration route, (2) determine the influence of the origin and release zone in the fishway location, and (3) define the main downstream moving periods. To achieve this, fishway location and passage success were monitored using tagged fish for two years. The delivered results improve the understanding of migration routes and raise awareness of the possible use of fishways as bidirectional connectivity devices.

## 2. Materials and Methods

### 2.1. Study Site

The study was carried out in the mainstem of the Duero River, near the village of Guma (Burgos), northwest part of Spain (Figure 1). The Duero River basin presents a high degree of fragmentation, with nearly 5200 obstacles in the Spanish part of the basin alone, from which, 141 are small HPPs and 23 are large HPPs [43,47].

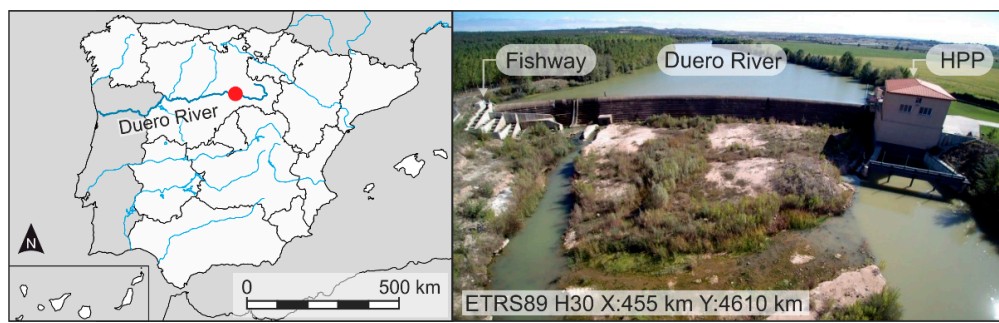

**Figure 1.** Location of Guma hydropower plants (HPP) and associated facilities.

The hydrology of the Duero River in the study river section is characterized by low flows in summer (5 $m^3$/s; exacerbated by upstream water diversions for irrigation) and medium–high flows during winter and early spring, associated with the rainy season and snow-melting episodes (30–45 $m^3$/s). The mean annual flow is about 17.6 $m^3$/s.

The river reach is dominated by small and medium-sized potamodromous rheophilic endemic cyprinids, such as the barbel, northern straight-mouth nase (*Pseudochondrostoma duriense* Coelho, 1985), northern Iberian chub (*Squalius carolitertii* Doadrio, 1987), and Pyrenean gudgeon (*Gobio lozanoi* Doadrio and Madeira, 2004). Their populations are currently suffering an important decrease (specially nase, categorized as endangered [48]), whereas the presence of alien species is increasing (mainly bleak *Alburnus alburnus* Linnaeus, 1758). River discharge during cyprinid upstream migration (April–July) is between 7 and 30 $m^3$/s. The river reach belongs to the Epipotamon zone [49] with an average altitude of around 810 m a.s.l. and corresponds to the C6 Rosgen category [50]. There are two unsurpassable dams 15.8 km downstream and 4.1 km upstream, respectively, of the studied river section (both have a fishway, although its efficiency is nil) which supposes the limits for fish movement.

The Guma HPP is a run-of-the-river HPP with a total height of 8.85 m, an installed capacity of 2.25 MW, and two Kaplan turbines. Due to the run-of-the-river configuration, there is no legal requirement for maintaining a minimum environmental flow through the dam. In the right bank of the dam, there is a pool and weir type fishway (Figure 2) composed of 36 cross-walls with submerged notches and bottom orifices (notch width = 0.3 m; sill height = 0.8 m; orifice size = 0.175 m × 0.175 m) and 35 pools (length = 2.6 m; width = 1.6; depth 1.2 m; slope = 8.6%), with mean water drops of 0.25 and a volumetric power dissipation of 121 ± 10 W/$m^3$. All the hydraulic parameters of the fishway are within the recommended ranges according to the most important design guidelines for this fishway type and target species [24,51,52]. Moreover, this fishway has been previously evaluated for

barbel upstream movements, showing a passage success of between 73 and 94% (confined and free trials; [53]). The fishway discharge operation varies from 0.25 to 0.50 m$^3$/s in order to ensure the correct operation of the fishway and that fish can correctly locate the downstream by-pass access from the river to the fishway. Likewise, the facility has an additional channel that joins the turbines outlet and the entrance bypass access, which provides a supplementary attraction flow near the fishway entrance (5% of the competing flow between the fishway and the turbines).

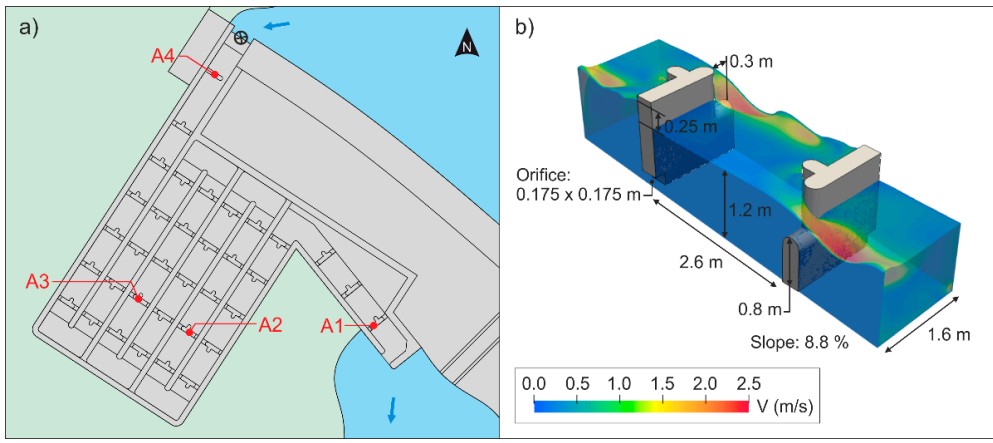

**Figure 2.** (**a**) Plan view of Guma fishway and passive integrated transponder (PIT) antenna location (A). (**b**) 3D scheme of the submerged notch with bottom orifice fishway together with principal hydraulic variables.

### 2.2. Fish Collection and Tagging

Fish collections were carried out during summer and autumn 2018 and 2019, with a variable frequency. Fish were captured by electrofishing (Hans-Grassl ELT60II backpack equipment; 180–250 V and 1.5–2.5 A) and/or by landing nets. Barbels were captured in two different locations, one into the Guma HPP fishway and near the downstream part of the fishway, and the second one 4 km downstream of the Guma HPP, near the following HPP in the same river (Vadocondes). Thus, fish were grouped depending on their origin: (1) fish captured at Guma were considered presumably acquainted with the fishway (hereinafter experienced fish); (2) fish captured at Vadocondes, presumably without experience of the Guma river section (hereinafter naive fish) (Table 1).

**Table 1.** Fish tagging samples. Experience: related to the fish origin. Release bank: release location of the riverbank side (700 m upstream Guma fishway). *n*: number of tagged barbels. SD: Standard Deviation. Range: minimum and maximum fork length (mm).

| Experience | Release Bank | *n* | Fork Length (mm) | |
| --- | --- | --- | --- | --- |
| | | | Mean ± SD | Range |
| Experienced | Left | 104 | 149 ± 8 | 91–293 |
| | Right | 81 | 144 ± 7 | 100–245 |
| Naive | Left | 38 | 139 ± 13 | 80–235 |
| | Right | 76 | 155 ± 15 | 77–420 |

All fish were anesthetized with eugenol (50 mg/L diluted in ethanol in the proportion 1:10), measured (fork length (FL), ±0.1 cm), weighted (±1 g) and tagged intraperitoneally with a half-duplex (HDX) passive integrated transponder (PIT) tag by an incision posterior to the left pectoral fin. Tags measured 12 or 23 mm long by 2.12 or 3.65 mm diameter and 0.1 or 0.6 g (in all cases respecting the relationship of tag weight lower than 2% of fish weight). After tagging, fish could recover for at least two hours before their release into the river. They were held in recovery boxes at ambient temperatures and with water supplied

directly from the river. After visual evidence of recovery from anesthetic effects, fish were released in two groups corresponding to the release riverbanks of the reservoir (i.e., right and left banks) (Table 1, Figure 2) approximately 700 m upstream of the Guma fishway.

In addition, during field visits in summer and mid-autumn, a large number of fish were always observed in the fishway and, to clarify their movement direction (resident, descent migrants, or ascent migrants), a complementary experiment was carried out between June and October 2020 (monthly frequency). This simple experiment consisted of the complete emptying of the fishway (i.e., all fish inside the fishway were captured and then released in the downstream section outside the fishway), followed by the installation of a close mesh halfway up the fishway. After 24 h, all fish that were found in the fishway were captured and categorized into two groups: (1) descent migrants, if they were found above the close mesh, i.e., in the upper part of the fishway; (2) ascent migrants if they were found below the close mesh, i.e., in the lower part of the fishway. It is worth mentioning that possible resident fish were classified as ascent migrants.

All experiments were performed following the European Union ethical guidelines (Directive 2010/63/UE) and Spanish Act RD 53/2013, with the approval of the competent authorities (Regional Government on Natural Resources and Water Management Authority).

### 2.3. Data Collection

Fish movements were monitored through PIT tag technology. This technology consists of a set of pass-through antennas that cover both the notch and the orifice of fishway crosswalls and records the passage of a PIT-tagged fish. Each antenna was connected to a reader (HDX multiplexer reader, ORFID®, Portland, OR, USA) programmed to send and receive information at 14 Hz (3.5 Hz or 0.29 s per antenna). Four antennas were placed into the fishway, two located at the most downstream (antenna 1) and upstream (antenna 4) pools, and the other two (antennas 2 and 3) halfway up the fishway. For antenna data processing, several assumptions and metric-definitions were made:

(1) For each fish, only records of the same year of the release date were considered (as the experiments lasted two years, recaptures in different years were possible).

(2) For each fish, a fishway location occurred if the fish was recorded first, and at least once in the uppermost antenna (antenna 4). If there was more than one location registered by fish, only the earliest one was considered.

(3) Location time of the fishway was defined as the time between the release and the first detection in the uppermost antenna (antenna 4).

(4) For each fish, a fishway entry occurred if the fish was recorded first in the uppermost antenna (antenna 4) and then in antenna 3 or 2. If there was more than one entry by fish, only the first one was considered.

(5) A success in the fishway downstream passage was defined for fish with a first record in antenna 4, a last record in antenna 1, and at least one intermediate record in antennas 3 and/or 2.

(6) The downstream transit time was only calculated for the fish with a successful downstream passage. If there were more than one success by fish, the one with the lowest downstream transit time was considered.

(7) Downstream transit time was defined as the time between the last detection in the uppermost antenna (antenna 4) and the first detection in the lowermost antenna (antenna 1).

(8) As complementary information, downstream passage through turbines or spillways was assigned to the fish with (1) first records in the lowermost antenna (i.e., they passed downstream through the turbines or spillways, and they came back to the fishway for upstream migration), or (2) first records in the uppermost antenna and subsequent records in lowermost antenna with an elapsed time between them of at least one day (i.e., fish tried to enter the fishway but exited, then passed through turbines or spillways, and then, came back to the fishway for upstream migration).

### 2.4. Hydraulic Characterization

To characterize the upstream region of Guma HPP and to compare the fish movements with the discharge and velocity distribution of the intake area of the turbines and the associated intake of the fishway, a set of bathymetries were conducted (Figure 3). All bathymetries were performed with an acoustic Doppler current profiler boat (OTT Q Liner 2 ADCP Boat). Next, all bathymetries together with on-site velocity measurements (Swoffer Model 2100 Current Velocity Meter) in the fishway and HPP intakes were combined to develop a surface contour plot (depth ≤1 m) of the study area.

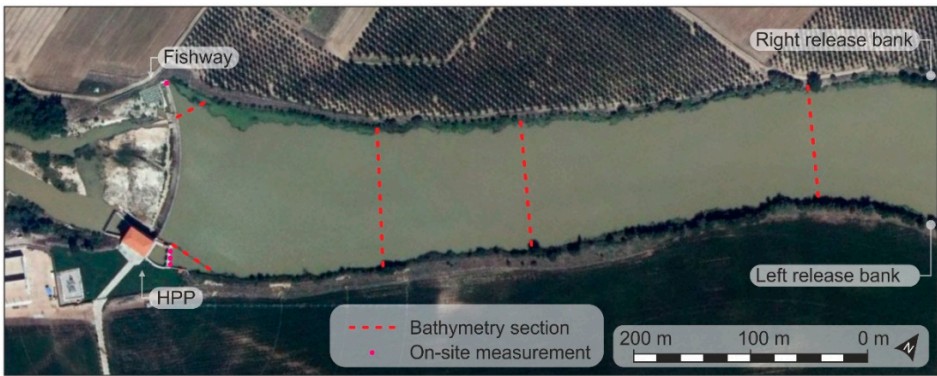

**Figure 3.** Studied sections, on-site velocity measurements, and release sites.

The combination of data in a single contour map was performed on a five-step procedure: (1) calculation of mean velocity up to 1.00 m of depth in each measured point, (2) peak removal by average filtering for each bathymetry section, (3) combination in Autodesk AutoCAD® of all measures in a single spatially referenced dataset, (4) a low-density linear interpolation in Autodesk AutoCAD® and (5) a high-density interpolation using the triangulated natural neighbor interpolation method from Matlab R2019a.

### 2.5. Data Analysis

All data analyses were performed in Statgraphics Centurion statistical software (Statgraphics Technologies, Inc., The Plains, VA, USA; Version 18.1).

To check possible differences between fish origin/experience (experienced and naive) and release bank (left and right) on the metrics of fishway location, entry, and downstream passage, the chi-square test of independence was used. In addition, the Mann–Whitney (MW) test was used to evaluate if there were any significant differences in location time and downstream transit time by origin/experience and release bank. Additionally, the MW test was used to evaluate the effect of fork length in the downstream passage proportion.

To identify major downstream migration periods, a frequency analysis of the number of fishway locations and downstream success was performed. Additionally, to detect a possible influence of day-hours on downstream movement, an hourly frequency analysis (GMT +01:00) was conducted, considering possible differences by origin/experience and release bank. Additionally, frequency analysis of captures during the period summer–autumn 2020 was carried out to identify the proportion of fish demonstrating downstream movement.

## 3. Results

Nearly a quarter of the tagged barbels (*n* = 74) were able to descend through the fishway, whereas only 26 were identified passing downstream by other routes (i.e., first recorded by the most downstream antenna). The hydraulic characterization of the river section allowed us to identify the area of the turbine intake as the most probable attractive route for downstream migration, as it provided a larger area of high flows than the intake of the fishway (Figure 4). Overflows by the spillways rarely occurred during the study period, only registering four possible barbels (of those 26) descending on dates in which sufficient water depth conditions occurred in the spillway.

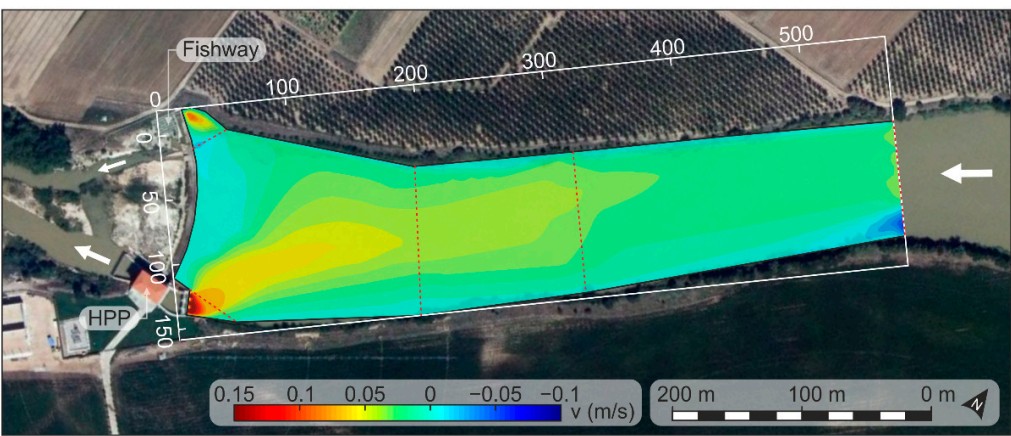

**Figure 4.** 2D velocity contour in the upstream area of the facility (discharge: 9.71 m$^3$/s).

Downstream metrics are summarized in Table 2. On the one hand, nearly 40% of tagged barbels were able to locate and approach the fishway, with an overall median time of approximately 2 days. From said barbels, 85.8% (34.5% of the total) were able to enter the fishway and 61.7% (24.9% of the total) completely descended the fishway. Experienced fish showed significantly greater proportions in fishway location, entrance and passage success metrics than naive fish, although without differences in median location time. As regards the release bank, fish released in the right bank showed significantly greater proportions of fishway location and entry, but with no significant differences for passage success and location time. Experienced barbels that were released on the right bank showed the best ratings, with a 70.4% location success rate and within these, a 66.7% downstream success rate (46.9% of the total). Meanwhile, the naive barbels released on the left bank offered the worst ratings, with a 26.6% location success rate and within these, a 50% downstream success rate (13.2% of the total).

On the other hand, fork length was found to significantly affect the downstream success rate (*p*-value = 0.002), larger barbels were more successful (median FL = 146 mm; *n* = 74) than smaller barbels (median FL = 132 mm; *n* = 225).

Downstream transit time lasted, in general, less than 10 h with a median value of 2.2 h, although with significant differences depending on the release bank (Table 3), with lower transit times for barbels that were released on the left bank. The origin/experience did not show a significant effect on the transit time, however, when considering its cross-effect with the release bank, the experienced barbels showed a significantly lower time if released on the left bank (*p*-value = 0.015), whereas the naive barbels did not show differences by release bank (*p*-value = 0.418).

**Table 2.** Proportion of fishway location, entry and downstream success, as well as median time to locate the fishway after release (in days) by origin/experience and release bank. $p$ corresponds to the $p$-value of the tests for the comparisons of proportions (chi-square test of independence) or time (Mann–Whitney test) between groups. n: number of fish.

| Metric | Global | Experience | | | | | | Release Bank | | | | | |
|---|---|---|---|---|---|---|---|---|---|---|---|---|---|
| | | Experienced | | | Naive | | | Left | | | Right | | |
| | | Release Bank | | Global | Global | Release Bank | | Experience | | Global | Global | Experience | |
| | | Left | Right | | | Left | Right | Experienced | Naive | | | Experienced | Naive |
| Location | 40.1% (120/299) | 34.6% (36/104) | 70.4% (57/81) | 50.3% (93/185) | 23.7% (27/114) | 26.3% (10/38) | 22.4% (17/76) | 34.6% (36/104) | 26.3% (10/38) | 32.4% (46/142) | 47.1% (74/157) | 70.4% (57/81) | 22.4% (17/76) |
| | | $p < 0.001$ | | $p < 0.001$ | | $p = 0.640$ | | $p = 0.350$ | | $p = 0.009$ | | $p < 0.001$ | |
| Entry * | 34.5% (103/299) | 29.8% (31/104) | 67.9% (55/81) | 46.5% (86/185) | 14.9% (17/114) | 21.1% (8/38) | 11.8% (9/76) | 29.8% (31/104) | 21.1% (8/38) | 27.5% (39/142) | 40.8% (64/157) | 67.9% (55/81) | 11.8% (9/76) |
| | | $p < 0.001$ | | $p < 0.001$ | | $p = 0.193$ | | $p = 0.301$ | | $p = 0.016$ | | $p < 0.001$ | |
| Downstream passage success | 24.8% (74/299) | 23.1% (24/104) | 46.9% (38/81) | 33.5% (62/185) | 10.5% (12/114) | 13.2% (5/38) | 9.2% (7/76) | 23.1% (24/104) | 13.2% (5/38) | 20.4% (29/142) | 28.7% (45/157) | 46.9% (38/81) | 9.2% (7/76) |
| | | $p < 0.001$ | | $p < 0.001$ | | $p = 0.517$ | | $p = 0.194$ | | $p = 0.099$ | | $p < 0.001$ | |
| Location time (median) | 1.9 days (n = 120) | 1.9 days (n = 36) | 1.8 days (n = 57) | 1.9 days (n = 93) | 2.0 days (n = 27) | 2.3 days (n = 10) | 2.0 days (n = 17) | 1.9 days (n = 36) | 2.3 days (n = 10) | 1.9 days (n = 46) | 1.8 days (n = 74) | 1.8 days (n = 57) | 2.0 days (n = 17) |
| | | $p = 0.303$ | | $p = 0.843$ | | $p = 0.744$ | | $p = 0.947$ | | $p = 0.267$ | | $p = 0.898$ | |

* Entry proportion could be underestimated since barbels could be recorded at the most upstream antenna (4), entered the fishway but did not reach the antenna 3 due to the distance between them (15 pools).

**Table 3.** Downstream transit time (in hours). *p* corresponds to the *p*-values of the Mann–Whitney test for the comparison of downstream transit time between the origin/experience and release bank categories. IQR: Interquartile range. *n*: number of fish.

| | | Global | Experience | | Release Bank | |
|---|---|---|---|---|---|---|
| | | | Experienced | Naive | Left | Right |
| Downstream transit time | Median | 2.2 h (*n* = 74) | 2.2 h (n = 62) | 3.2 h (n = 12) | 1.5 h (*n* = 29) | 4.8 h (*n* = 45) |
| | IQR | 1.0–8.7 h | 1.0–7.8 h | 0.8–9.3 h | 0.7–3.1 h | 1.2–9.9 h |
| | *p*-value | | *p* = 0.988 | | *p* = 0.011 | |

The downstream movements of barbel were recorded during most of the year except for winter, with the highest number of records between July and September (both included) (Figure 5). Complementary observations during June–October 2020 showed that, within this period, most of the barbel displacements corresponded with downstream movements (Table 4). Additionally, it was observed that most of the movements of the tagged barbels occurred during the daylight hours, with peak records during sunrise, midday and sunset (Figure 6).

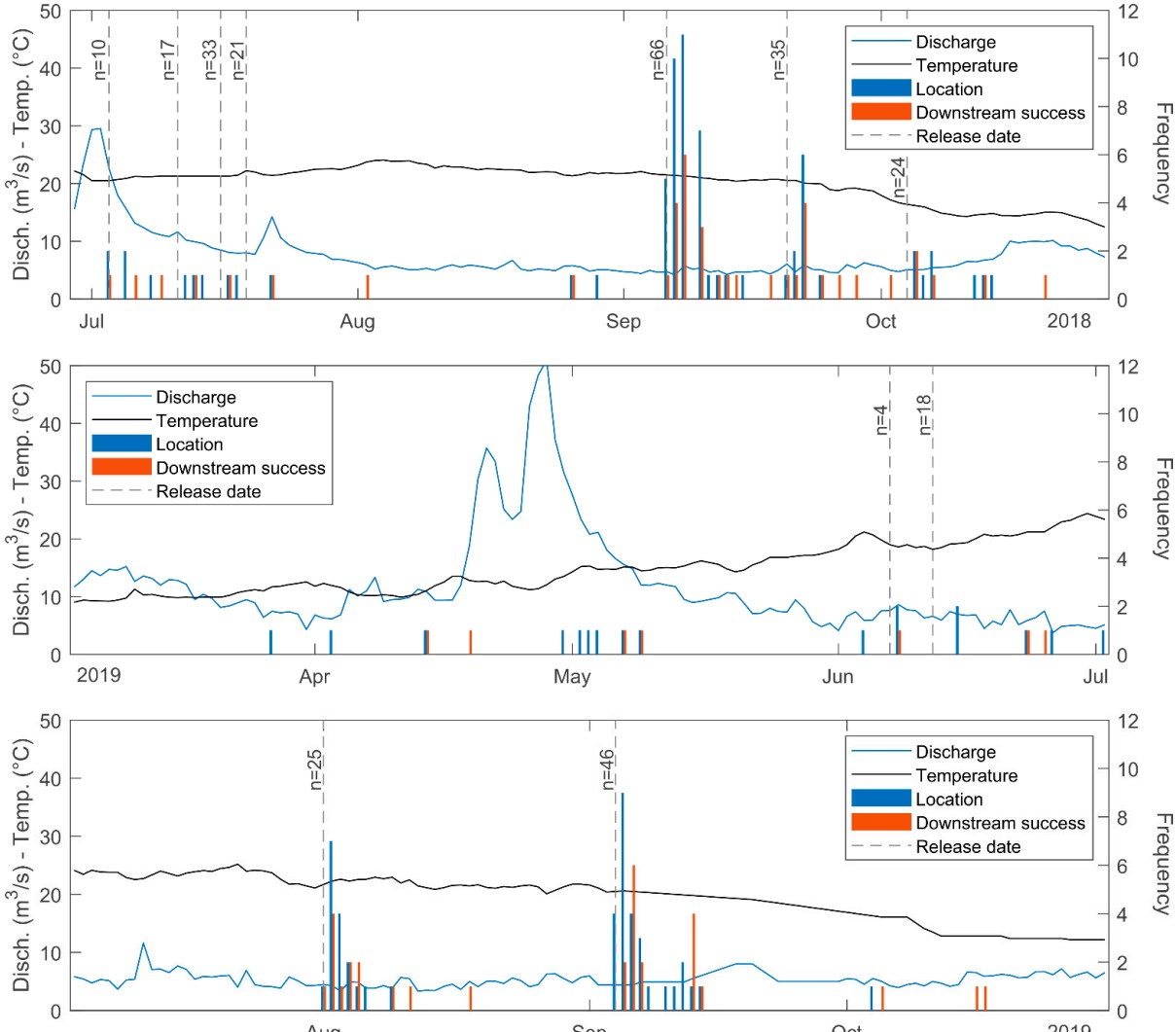

**Figure 5.** Location and downstream success, related to river discharge and water temperature. The logging system was running continuously from 01/05/2018 to 31/12/2019, however, periods without fish records were excluded from the illustration for clarity reasons.

**Table 4.** Proportion of barbel moving downstream considering the total sampled in the fishway. Between brackets (number of barbel descending/sum of barbel descending and ascending).

| Month (2020) | June | July | August | September | October |
|---|---|---|---|---|---|
| Downstream proportion | 58% (11/19) | 71% (12/17) | 35% (6/17) | 50% (21/42) | 100% (3/3) |

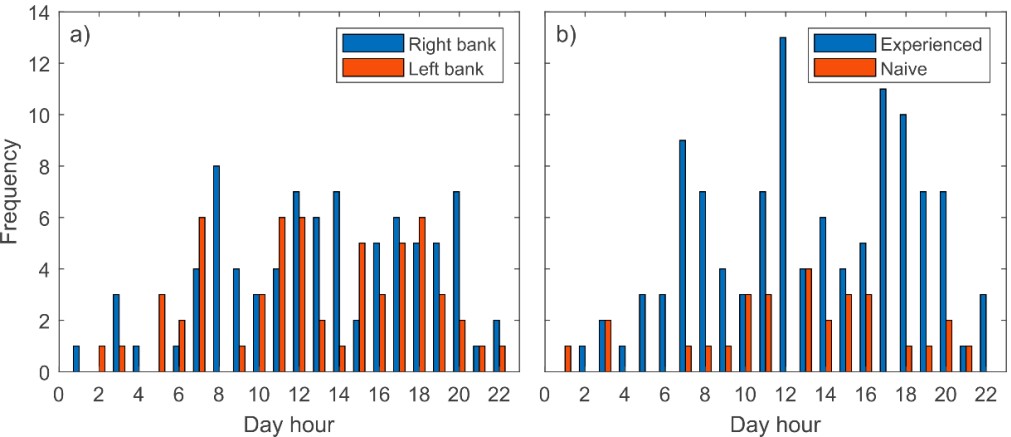

**Figure 6.** Location frequency at different hours in the day (GMT +01:00). (**a**) Considering the release bank. (**b**) Considering fish origin/experience.

## 4. Discussion

Most of the research resources and efforts invested in the study of the effectiveness of fishways have been focused on upstream migration, while research studies on evaluating the potential of these devices as possible routes for downstream migration are still scarce and, in general, it is assumed that their viability is minimal [30,37,38]. In many cases, such studies were conducted in large river dams, where the fishway dimension was negligible in respect to turbines and spillways [36,54]. However, even so, some encouraging results can be found for both large [39] and small dams [55]. Therefore, it seems that this general assumption may not be right. Being a site/flow-dependent situation, that is to say, it seems that the viability of fishways as a downstream migration route depends on the dam, the HPP operation and/or configuration, fishway placement, as well as on the relative discharge of the fishway in comparison with other possible routes.

In this particular case, considering the general configuration of the studied facility, with a distance of 135 m from the fishway to the HPP intake and an average ratio of discharge between the fishway and the HPP intake of 5%, a low use of the fishway for downstream migration was expected. However, the results show that downstream movements were noticeable. A total of 40.1% of fish located the fishway in the same season they were released and more than 61.7% of them used this route to move downwards (24.8% of total fish). Moreover, considering that the tagged fish, depending on their motivation, could move upstream, stay in the reservoir or go downstream, the real downstream passage success (i.e., the percentage of fish that really wanted to move downwards) could be greater. Additionally, some descending fish seemed to use the fishway as a summer habitat and they did not contribute to the passage success metric (as they did not exit the fishway). Thus, downstream fishway global efficiency could also be underestimated. In this sense, also taking into consideration the results from the complementary experiments, where approximately half of the captured barbels inside the fishway were moving downward, it is possible to conclude that a fishway may serve as a complementary route both in the study site as well as on other run-of-the-river small HPPs with similar characteristics.

Nevertheless, it is worth mentioning that the Guma HPP makes use of all the river discharge except for the proportion flowing through the fishway, thus, the spillway is only a downstream migration option under peak-flow events (winter season [44]). According to

the results, the downstream movement of fish occurred during the whole year, except for the wintertime, reaching the maxima between June and September during daylight hours (mainly from 7:00 to 20:00 h GTM+1). Therefore, as the downstream spillway route is only available in winter, when cyprinids are overwintering and show very low activity [2], the fishway usage probability increases during the rest of the year.

A greater proportion of fish collected in the Guma fishway and surrounding area (experienced fish) were shown to have located and entered the fishway when compared with those captured at 4 km downstream (naive fish). Since captured fish could have been resident, ascending, or descending fish, the resident and descending experienced fish could have explored the reservoir beforehand, and consequently, their probability of finding the fishway could be greater. In contrast, naive fish had a lower probability of being previously inside the fishway and the reservoir area beforehand. Therefore, considering that real descending fish should have acquired experience of the reservoir and the fishway when performing upstream spawning migration, experienced fish results may be closer to reality. The importance of fish origin has been highlighted in previous field work [56,57], thus, special attention has to be paid to the origin when conducting this type of fishway assessment and interpreting their results. Furthermore, the size of fish also has to be taken into consideration, as the present study shows that large fish were more successful in descending the fishway, probably due to higher levels of motivation. These results are in accordance with the results by Havn et al. [55].

Rheophilic species, such as barbels, tend to move downstream following the highest flow volume areas [28,31], which are characterized by the highest water velocities but the lowest velocity gradients. Furthermore, non-salmonids species generally prefer surface waters [29] and shorelines [31]. In the studied site, the highest flow volume area is visible from the water intake of the HPP up to 150 m upstream. In this area, the main flow volume travels from the middle of the cross-section to the powerhouse intake. Fish were released 700 m upstream of the dam, a site without a clear pattern in the flow towards the HPP intake. This suggests that fish released in the right-side bank could have followed the right shoreline, not detecting the main waterway, being able to locate the entrance of the fishway in a higher proportion. Likewise, the fish origin showed a clear cross effect with the release place. a higher proportion of experienced fish than naive fish released at the right bank located the fishway, whereas both experienced and naive fish had a poor success rate in locating the entrance when released at the left bank. This may indicate that the fish with real experience in the area may have only previously explored the right side (for example, by continuing near the shoreline when exiting the fishway during the upstream migration), behaving like inexperienced fish when released at the left bank. The return of fish to previously occupied places is related to the topographic mid-term memory, involving the visual or olfactory identification of landmarks and surface topography [2,58]. Therefore, a fish may choose a route for downstream migration by considering its previous experiences.

Despite their effect on fishway location, both the release bank and the fish experience did not influence the fishway location time. Fish took about 2 days to locate the entrance of the fishway. This location time may be considered fast when compared with location times for barbel during upstream movements (5–12 days [57,59]). This corroborates the possible previous experience acquired by the fish, which may have facilitated locating the fishway entrance without exploring other areas.

As regards the descent through the fishway, fish expended a median transit time of around 2.2 h; a lower value than ascent transit time for the same fishway (median = 3.5 h and IQR = 2.4–45.7 h [60]) and this seems to suggest a non-relevant downstream migration delay. Experienced fish released at the left bank descended faster than those released at the right bank (the same as the fishway). This result may indicate that the low proportion of experienced fish from the left margin which managed to cross the reservoir and located the fishway were probably the most motivated and skilled individuals.

When downstream migration occurred through the turbine route, the fish that reach the turbine intake following the bulk flow may experience a rapid change in water velocity

and depth, especially if they are near the surface, which may stop or delay the passage [26]. Once the fish approaches the HPP intake, the location of alternative routes is quite difficult (due to the surrounding hydrodynamics), thus, even if passage through deep turbines is not their preferable route, it may be a passage of last resort [31]. At least 26 barbels (8% of sampled fish) moved downwards through the turbines (probably the majority) or spillway (a few of them, only during high flow events). Considering the experimental design and the complementary nature of this result, it is very probable that the number of barbels selecting these routes was higher. Previous experiences with salmonids in small HPP dams with a strong bulk flow to the turbines and spillway, showed a clear preference for spillway routes, especially for juveniles [32,61]. Havn et al. [55], in a small HPP with two separately groups of turbines, detected a descent passage ratio of 19% and 25% through the two associated fishways (vs. 81% and 75% through turbines), with competing flows of about 1.5 and 8%, respectively. Celestino et al. [39], in a large neotropical dam, revealed a 14% efficiency for downstream migration of a fishway with an entrance located close to the turbines and the shoreline, with a competing discharge of 0.1%. Taking this into account, the correct placement of the fishway (e.g., near the turbine intakes and shoreline) in combination with specific downstream facilities such as sluices or guiding-bypass systems, could increase their usage probability. Another possibility to attract fish to the fishway would be to increase the flow in the surrounding area of the fishway. However, this would reduce the discharge through turbines and, consequently, the power plant profit.

This study shows that downstream movements of potamodromous fish occurred almost throughout the whole year, with an increase in records during summer and autumn, probably related to post-spawning and summer refuge movements [5,10–12], as well as movements to search for overwinter places, respectively [14,15]. Thus, downstream movements must be considered in any connectivity restoration projects, as their importance for fish is comparable to upstream movement. To survive and complete their life cycle, fish need to move, both upstream and downstream. Fishways could also be a route for downstream migration, but are not a complete solution by themselves. The correct placement and dimensioning of the fishway, supported by guidance structures and attractive flows, could improve its descent efficiency, turning fishways into a bidirectional migration facility. However, further research efforts and resources are necessary to achieve this. Among others, it is necessary to improve the knowledge on relationships between fish behavior and their ability to locate and use downstream bypasses. Taking this into consideration, the results of this study may be used to help operators, engineers and biologists turn HPPs into environmentally friendly energy production facilities.

### 5. Summary and Conclusions

- A considerable percentage of barbel used the fishway for downstream movements, despite the low ratio between discharge and velocity fields at the turbines and the fishway, as well as the great distance between them.
- The distribution of fish along the cross-section affects the selection of passage routes in reservoirs with low water velocity and without clear waterways up to the turbine intakes.
- Previous experience of barbels affects their ability to locate the fishway.
- Barbels perform downstream movements from spring to autumn, and mainly during daylight.
- Fishways may serve as a complementary route in run-of-the-river small HPP with similar characteristics to the study site.

**Author Contributions:** Conceptualization, F.J.S.-R. and F.J.B.-C.; methodology, F.J.S.-R. and F.J.B.-C.; validation, J.F.F.-P. and A.G.-V.; field experiments F.J.S.-R., F.J.B.-C. and A.G.-V.; data curation, F.J.B.-C.; writing—original draft preparation, F.J.S.-R.; writing—review and editing, J.F.F.-P. and A.G.-V.; project administration, F.J.B.-C.; funding acquisition, F.J.S.-R. All authors have read and agreed to the published version of the manuscript.

**Funding:** This project has received funding from the European Union's H2020 research and innovation program under grant agreement No. 727830, FIThydro. Contributions of Juan Francisco Fuentes-Pérez and Francisco Javier Bravo Cordoba were partly financed by a Torres Quevedo grant PTQ2018-010162 and PTQ2016-08494, respectively. Ana García-Vega's contribution was financed by a Ph.D. grant from the University of Valladolid PIF-UVa 2017.

**Institutional Review Board Statement:** The study was conducted according to the European Union ethical guidelines (Directive 2010/63/UE) and Spanish Act RD 53/2013 for the protection of animals used for scientific purposes, with the favorable report from the Animal Research Ethics Committee from the University of Valladolid as well as the approval of the competent authorities, i.e., Regional Government on Natural Resources (Junta de Castilla y León) and Water Management Authority (Confederación Hidrográfica del Duero).

**Acknowledgments:** We specifically thank Juan Carlos Romeral de la Puente (SAVASA) for letting us use his installations in Guma HPP. We would also like to thank the GEA staff and the Itagra.ct technological center for logistical support, as well as fishing services of Regional Government (Castilla y León) and Water Management Authority (Confederación Hidrográfica del Duero).

**Conflicts of Interest:** The authors declare no conflict of interest.

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
