# Peer review of "Fishways as Downstream Routes in Small Hydropower Plants: Experiences with a Potamodromous Cyprinid"

_water, doi:10.3390/w13081041_

Round 1
Reviewer 1 Report
The authors describe the migration of fish downstream and through the hydropower plant. The migration of fish down the river bed and the crossing of water structures from the top down by fish is an innovative and little known topic. Typically, the upstream migration of fish and the upward movement of structures are studied. The research was planned and performed correctly. They consisted of field measurements (bathymetry, ichthyology) and study analyzes (statistical tests, analyzes). The quality of field measurements and their subsequent analyzes do not raise any doubts. The results and conclusions result from the course of the conducted research and are related to them. The article is interesting because it deals with the rarely described phenomenon of fish migration down the watercourse. I have one remark: Why did you choose this hydraulic structure? Did it have any unusual solutions? I would also add a scheme of the fish pass with dimensions and construction details.
Author Response
Please, see the attached file.

Reviewer 2 Report
The article is very interesting. I have provided my comments below.
Please add a detailed ichthyological analysis of fish moving through the river.
Does the fishway function properly, are the minimum depths of water in chambers, slots ensured?
Do the water velocities in the fish ladder not exceed the acceptable values?How much is the outflow of an additional fish attracting channel?
What solutions can be applied to attract fish towards the fish ladder and not towards the power plant?
Is the continuity of the river ensured along the whole length?
Please detail your summary.
Author Response
Please, see the attached file.

Reviewer 3 Report
Dear Authors,
I am writing this to submit my comments on your research article with the following details.
Manuscript title: Fishways as downstream routes in small hydropower plants: experiences with a potamodromous cyprinid
Manuscript Number: water-1159359
Journal Submitted: Water
General Comments:
The manuscript has several basic-level English language and grammar errors that can be fixed with an English language expert's help.
Specific Comments:
Title:
The title is self-explanatory.
Abstract:
The abstract is not abstract; instead, you are recreating some parts, especially why you took this study and what you aimed for, along with a simple conclusion which is already evident from your title.
This is not acceptable. Please revise it as a whole and remove the objectives mentioned—no need to put long details of this study's basics. Instead, tell us which methods you have used and what results were obtained with enriched conclusions of this study.
Keywords: Please add more and relevant.
Introduction:
4th paragraph: At the end, you have mentioned there are some conflicting studies. Please highlight the main points.
Please delete or shift the sentences mentioned after the objectives.
This is a well-written part.
Materials and Methods:
Please describe the fish collection methods.
The rest is done excellently.
Results:
The results should be extended in their description to reveal more details.
Discussion:
The first paragraph is much more like recreating the details already given in the introduction. Please delete from here or from the introduction section.
The rest is fine.
Figures and Tables:
All figures and tables are excellent and necessary.
Conclusions
No provided. Please designate the last paragraph of the discussions as conclusions and extend it with more insights.
References:
Fine.
Author Response
Please, see the attached file.

Round 2
Reviewer 3 Report
No more corrections required from my side.